# Analysis of *MC1R, MITF, TYR, TYRP1*, and *MLPH* Genes Polymorphism in Four Rabbit Breeds with Different Coat Colors

**DOI:** 10.3390/ani11010081

**Published:** 2021-01-05

**Authors:** Xianbo Jia, Peng Ding, Shiyi Chen, Shaokang Zhao, Jie Wang, Songjia Lai

**Affiliations:** Farm Animal Genetic Resources Exploration and Innovation Key Laboratory of Sichuan Province, Sichuan Agricultural University, Chengdu 611130, China; jaxb369@sicau.edu.cn (X.J.); m15008309752@163.com (P.D.); chensysau@163.com (S.C.); z13503321824@163.com (S.Z.); wjie68@163.com (J.W.)

**Keywords:** *MC1R*, *MITF*, *TYR*, *TYRP1*, *MLPH*, coat color, rabbit

## Abstract

**Simple Summary:**

Coat color is an important breed characteristic and economic trait for rabbits, and it is regulated by a few genes. In this study, the gene frequencies of some pigmentation genes were investigated in four Chinese native rabbit breeds with different coat colors. A total of 14 genetic variants were detected in the gene fragments of *MC1R, MITF, TYR, TYRP1*, and *MLPH* genes, and there was low-to-moderate polymorphism in the populations. The gene frequency showed significant differences among the four rabbit populations. The above results suggest that these genetic variations play an important role in regulating the coat color of rabbits. This study will provide potential molecular markers for the breeding of coat color traits in rabbits.

**Abstract:**

Pigmentation genes such as *MC1R, MITF, TYR, TYRP1*, and *MLPH* play a major role in rabbit coat color. To understand the genotypic profile underlying coat color in indigenous Chinese rabbit breeds, portions of the above-mentioned genes were amplified and variations in them were analyzed by DNA sequencing. Based on the analysis of 24 Tianfu black rabbits, 24 Sichuan white rabbits, 24 Sichuan gray rabbits, and 24 Fujian yellow rabbits, two indels in *MC1R*, three SNPs in *MITF*, five SNPs (single nucleotide polymorphisms) in *TYR*, one SNP in *TYRP1*, and three SNPs in *MLPH* were discovered. These variations have low-to-moderate polymorphism, and there are significant differences in their distribution among the different breeds (*p* < 0.05). These results provide more information regarding the genetic background of these native rabbit breeds and reveal their high-quality genetic resources.

## 1. Introduction

Coat color is one of the important characteristics of animals. It often reflects breed characteristics, production value, species purpose value, and economic value [1,2]. The diversity of animal coat color is regulated by several genes, and the different colors are usually regulated by the major genes. These genes not only affect the change of coat color after mutation, but also control the formation of coat color through interaction [3,4]. So far, variations in 378 genes thought to be related to coat color formation in mammals have been found, of which 45% have been cloned and identified; almost half of the proteins encoded by these genes are specific or non-specific to melanosomes [5].

The melanocortin 1 receptor (*MC1R*) gene is mainly expressed in hair follicles and skin melanocytes and is closely related to skin pigmentation. The *MC1R* complex signal can activate α melanocyte-stimulating hormone, which leads to the production of melanin and dark brown eumelanin. Three deletion sites in the coding region of the rabbit *MC1R* gene (c.(124A;125_130del6), c.280_285del6, c.304_333del30) were found to be associated with white, black, red, and yellow phenotypes in rabbits [6,7]. The microphthalmia-associated transcription factor (*MITF*) gene is a key regulator in melanin synthesis; melanin participates in the proliferation, differentiation, and transport of melanocytes [8]. After *MITF* silencing in human melanocytes, the expression of *TYR* and *TYRP1* protein decreased significantly at different levels, while the expression of *TYRP2* protein increased significantly. Human *MITF* variations lead to type II Waardenburg syndrome. Patients usually have symptoms such as congenital cataracts, skin hypopigmentation, and nervous deafness [9], while affected mice show micro-ocular malformations, early-onset deafness, and hypopigmentation of fur and irises [10]. Tyrosinase (*TYR*) is involved in the synthesis of dopa and dopaquinone in the process of melanin synthesis [11]. The type and quantity of melanin synthesis is determined by its activity and concentration. High *TYR* activity and concentration lead to the production of true melanin, otherwise brown melanin is produced [12]. The variation of the rabbit *TYR* gene (T373K) leads to alteration of the last N-glycosylation site of the *TYR* coding sequence, which is significantly related to rabbit albinism [13]. The tyrosinase-related protein (*TYRP1*) gene is a member of the tyrosinase-related protein family and is a key factor in melanin synthesis [11]. The nonsense variation (g.41360196G> A) of rabbit *TYRP1* results in the early termination codon at position 190 of the amino acid sequence (p.Trp190ter), which is significantly correlated with the brown phenotype of rabbits [14]. The melanophilin (*MLPH*) gene is mainly involved in the transport of mature melanosomes in melanocytes and makes melanosomes gather at the dendritic ends of melanocytes. When *MLPH* is overexpressed in melanocytes, it induces the aggregation of melanosomes to regulate animal coat color. Two variations in the coding region of the rabbit *MLPH* gene (c.111-5C > A, c.585delG) were found to be associated with color dilution of rabbit coat color [15].

There are more than 20 indigenous and imported rabbit breeds in China, which are bred for the production of their meat, fur, and wool [16]. The production performance of Chinese indigenous rabbit breeds is generally lower than that of imported breeds, but indigenous breeds have advantages in adaptability, disease resistance, meat quality, and coat color. Unfortunately, the genetic diversity and characteristic genes of Chinese indigenous rabbits have not been well studied yet. In this study, the polymorphisms of *MC1R*, *MITF, TYR, TYRP1*, and *MLPH* were obtained in four Chinese native rabbit breeds by DNA sequencing to analyze the relationship between these polymorphic sites and different coat color phenotypes. It may provide a basis for the mechanism of rabbit coat color formation and provide theoretical support for rabbit coat color breeding and genetic improvement in the future.

## 2. Materials and Methods

### 2.1. Ethics Statement

Collection of biological samples and experimental procedures involved in this study were approved by the Institutional Animal Care and Use Committee at the College of Animal Science and Technology, Sichuan Agricultural University, China (No. DKYB20190103).

### 2.2. Rabbit Sampling

Ear clips from 96 adult rabbits from four Chinese rabbit breeds were collected in this study, including Tianfu black rabbit (TB, 24), Sichuan white rabbit (SW, 24), Sichuan gray rabbit (SG, 24), and Fujian yellow rabbit (FY, 24) [17]. According to pedigree records, the genetic relationships within three generations were avoided for all animals. The genomic DNA was extracted using the TIANamp Genomic DNA kit (TianGen™, Beijing, China), according to the manufacturer’s instructions, and then stored at −20 °C for later analysis.

### 2.3. PCR Amplification and DNA Sequencing

The candidate variations were selected from former publications and subjected to genotyping in the present study [2,7,14,15,18]. Based upon the *Oryctolagus cuniculus’ MC1R*, *MITF*, *TYR*, *TYRP1*, and *MLPH* gene sequences (FN658678.1, NC_013677.1, NC_013669.1, NC_013669.1, and NW_003159466.1), five pairs of primer sequences (Table 1) were designed by the Primer Premier 5.0 software to amplify the target fragments. PCR was performed in 20 μL of reaction volume, containing 50 ng genomic DNA, 10 μM of each primer, 2 × MasterMix (0.05 units/mL Taq DNA polymerase, 4 mM MgCl_2_, 4 mM dNTPs), and double-distilled (dd)H2O (Aidlab Biotechnologies Co., Ltd., China). The PCR protocol was as follows: 95 °C for 5 min, 35 cycles of denaturing at 94 °C for 30 s, annealing at Tm °C (Table 1) for 30 s, and extension at 72 °C for 30 s, with a final extension at 72 °C for 10 min. The products for sequencing were first electrophoresed on 1.5% agarose gels with GoldView^TM^ nucleic acid stain (Solarbio, China), and then purified using an Axygen^TM^ kit (BMI Fermentas, Glen Burnie, MD, USA) and sequenced in both directions in an ABI PRIZM 377 DNA sequencer (Perkin-Elmer). The SeqMan software package was used to analyze the sequence maps.

### 2.4. Statistical Analysis

Genotype frequencies and allelic frequencies were determined by direct counting. The Hardy–Weinberg expectation was measured through a χ^2^ test. Population polymorphism information content (PIC) was calculated according to Nei’s methods [19]. Difference in the genotype distribution among breeds was assessed using a Fisher’s exact or chi-square test using the R language (version 3.6.1, http://www.R-project.org).

## 3. Results

### 3.1. SNPs of Five Gene Fragments in Four Rabbit Breeds

A 606 bp fragment of the *MC1R* gene, a 729 bp fragment of the *MITF* gene, a 741 bp fragment of the *TYR* gene, a 682 bp fragment of the *TYRP1* gene, and a 498 bp fragment of the *MLPH* gene were amplified by PCR. *MC1R, MITF*, *TYR*, *TYRP1*, and *MLPH* gene fragments from the four rabbit breeds were of the expected sizes.

In the *MC1R* gene fragment, two indels (c.284-285del, c.292-295del) were found in TBs and SWs. In the *MITF* gene fragment, three SNPs (g.232587 A < G, g.232650C < G, g.232766A < T) were found in TBs, SGs, TBs, and SWs. In the *TYR* gene fragment, five SNPs (c.185G < A, c.465C < T, c.498T < C, c.669C < T, c.624C < T) were found in TBs, whereas c.185G < A in SGs and c.624C < T in SWs and SGs were also observed. The *TYRP1* gene fragment revealed only one SNP (g.4137286G < A) in all four breeds. Three SNPs (c.693C < G, c.851A < G, c.911G < A) in fragments of the *MLPH* gene were only found in TBs and SWs. According to the number of SNPs in these five gene fragments, diversity was observed in the four rabbit breeds with the decreasing order of TB, SW, SG, and FY.

### 3.2. Genotype Frequencies of Five Gene Fragments in Four Rabbit Breeds

Allele and genotype frequencies were statistically analyzed for *MC1R*, *MITF*, *TYR*, *TYRP1*, and *MLPH* gene fragments, and the PIC values were calculated in each breed (Table 2, Table 3, Table 4, Table 5 and Table 6). The Fisher’s exact or χ^2^ test was used to compare the genotype frequencies between different breeds. For the *MC1R* gene fragment, two indels loci were found in TBs and SWs. These indels showed a low-to-moderate PIC value ranging from 0.1411 to 0.3750 with an average of 0.2949 in the two breeds. At c.284-285del, the normal type was equal to the indel type in TBs, while the normal type was most frequent in SWs at 75.00%. At c.292-295del, the normal type was the most frequent type in both breeds with 62.50% in TBs and 91.67% in SWs. Normal type and indel type distribution were not significantly different between the two breeds at the c.284-285del locus (*p* > 0.05, χ^2^ test), while the distribution of the two types were significantly different between the two breeds at the c.292-295del locus (*p* < 0.05, Fisher’s exact test).

For the *MITF* gene fragment, three SNPs were found in TBs, SGs, and SWs. These SNPs showed a moderate PIC value ranging from 0.2755 to 0.3750 with an average of 0.3253 in three breeds. At the g.232587 A < G locus, AG was the most frequent genotype in TBs with 41.67%, while AA was the most frequent genotype in SGs with 75.00%. Allele A was equal to G in TBs, whereas A was the most frequent allele in SGs with 79.17%. The genotype distribution was significantly different between the two breeds (*p* < 0.01, Fisher’s exact test). At the g.232650C < G locus, the three genotypes were equal to each other in TBs. At the g.232766A < T locus, genotype TT was the most frequent genotype with 70.82%, and allele T was the most frequent allele in SWs with 70.83%.

For the *TYR* gene fragment, five SNPs were found in TBs, SWs, and SGs. These SNPs showed a low-to-moderate PIC value ranging from 0.1948 to 0.3750 with an average of 0.3275 in three breeds. At the c.185G < A locus, GA was the most frequent genotype in TBs with 75.00%, while AA was the most frequent genotype in SGs with 58.33%. Allele A was the most frequent allele in TBs and SGs with 54.17% and 79.17%, respectively. The genotype distribution was significantly different between the two breeds (*p* < 0.01, Fisher’s exact test). At the c.465C < T locus, genotype CT was the most frequent genotype with 75.00%, and allele T was the most frequent allele with 54.17% in TBs. At the c.498T < C locus, genotype TC was the most frequent genotype with 78.26%, and allele T was the most frequent allele with 54.17% in TBs. At the c.669C < T locus, genotype CT was the most frequent genotype with 66.67%, and allele T was equal to C in TBs. At c.624C < T, AA was the most frequent genotype in TBs and SWs with 83.33% and 62.50%, respectively, while GG was the most frequent genotype in SGs with 79.17%. Allele A was the most frequent allele in TBs and SWs with 87.50% and 77.08%, respectively, whereas allele G was the most frequent allele in SGs with 81.25%. The genotype distribution was significantly different between TBs and SGs (*p* < 0.01, Fisher’s exact test) and between SWs and SGs (*p* < 0.01, Fisher’s exact test).

For the *TYRP1* gene fragment, only one SNP locus was found in all four breeds. The SNP showed a low-to-moderate PIC value ranging from 0.2392 to 0.3733 with an average of 0.3285 in four breeds. Genotype GA was the most frequent genotype in TBs, SGs, and FYs with 41.67%, 45.86%, and 62.50%, respectively, while AA was the most frequent genotype in SWs with 66.67%. Allele A was the most frequent allele in TB, SW, SG, FY with 54.17%, 83.33%, 60.42%, and 68.75%, respectively. The genotype distribution was significantly different between TBs and SWs (*p* < 0.01, Fisher’s exact test), between TBs and FYs (*p* < 0.05, Fisher’s exact test), and between SWs and SGs (*p* < 0.05, Fisher’s exact test).

For the *MLPH* gene fragment, three SNPs were found in TBs and SWs. These SNPs showed a moderate PIC value ranging from 0.3533 to 0.3750 with an average of 0.3668 in two breeds. At the c.693C < G locus, GG was the most frequent genotype in TBs with 62.50%, while CG was the most frequent genotype in SWs with 58.33%. Allele G was the most frequent allele in TBs with 64.58%, whereas C was most frequent allele in SWs with 54.17%. The genotype distribution was significantly different between the two breeds (*p* < 0.01, Fisher’s exact test). At the c.851A < G locus, GG was the most frequent genotype in TBs with 62.50%, while SWs did not have a most frequent genotype. Allele G was the most frequent allele in TBs with 64.58%, whereas allele C was equal to G in SWs. The genotype distribution was significantly different between the two breeds (*p* < 0.05, Fisher’s exact test). At the c.911G < A locus, AA was the most frequent genotype in TBs with 54.17%, while AG was the most frequent genotype in SWs with 41.67%. Allele G was the most frequent allele in TBs with 54.17%, whereas A was the most frequent allele in SWs with 54.17%. The genotype distribution was significantly different between the two breeds (*p* < 0.01, Fisher’s exact test).

### 3.3. Relationship between Coat Color and MC1R, MITF, TYR, TYRP1, and MLPH Gene Polymorphisms

There were significant effects of *MC1R*, *MITF*, *TYR*, *TYRP,1* and *MLPH* genotypes on coat color. Four SNPs may affect the black trait including c.465C < T, c.498T < C, and c.669C < T at the *TYR* gene and g.232650C < G at the *MITF* gene. SNP g.232766A < T at the *MLPH* gene may affect the white trait. Four variations may affect the black and white traits including c.292-295del at *MC1R* gene and c.693C < G, c.851A < G and c.911G < A at the *MLPH* gene. Two SNPs may affect the black and gray traits including c.185G < A at the *TYR* gene and g.232587A < G at the *MITF* gene. SNP c.624C < T at the *TYR* gene may affect the black, white, and gray traits. SNP g.4137286G < A at the *TYRP1* gene may affect the black, white, gray, and yellow traits.

## 4. Discussion

Coat color is an important characteristic in rabbit breeds, and domestic animal geneticists have always paid attention to its genetic mechanism [20]. Melanin and its derivatives are the main components of tyrosine-derived pigments, and their type and quantity determine the rabbit coat color [11]. The biosynthesis of melanin is a complex process, which requires the regulation of multiple signal molecules and specific components; the *MC1R*/cAMP signal pathway is one of the key components of this process. After binding to α-MSH, *MC1R* induces an increase in cAMP by the action of G protein. Then, cAMP exerts its signal molecule function through PKA to upregulate *MITF* gene expression. *MITF* upregulates the expression of melanin-related genes *TYR*, *TYRP1*, and *TYRP2* to catalyze the formation of melanin by L-tyrosine [5,21]. *MLPH*, Rab27a, and Myo5a form ternary complexes, which play a key role in the transfer of melanoma bodies from melanocytes to neighboring keratinocytes and ultimately color the animal coat [4,22]. Here, we investigated the variations of *MC1R*, *MITF*, *TYR*, *TYRP1*, and *MLPH* gene fragments in 4 rabbit breeds of different colors, and found 2 indels and 12 SNPs in these 5 gene fragments. The polymorphism distribution levels of the five genes were significantly different among these four rabbit breeds. These polymorphisms may be involved in the formation of rabbit coat color.

Genetic diversity is an important basis for evaluating the status of breeding germplasm resources, and it is the genetic basis for the population’s adaptation to the environment and evolution. The PIC is an important indicator reflecting the population’s genetic diversity. Usually, a PIC < 0.25 indicates low polymorphism, 0.25 < PIC < 0.5 indicates moderate polymorphism, and PIC > 0.5 indicates high polymorphism [23]. For the five gene fragments in this study, moderate polymorphism was observed at all SNPs in the breeds with polymorphism, except that low polymorphism was observed at *MC1R* c.292-295del in SWs, *TYRP1* g.4137286G < A in SWs and *TYR* c.624C < T in TBs. Moderate polymorphism was observed at 13 loci in TBs, 6 loci in SWs, 4 loci in SGs, and 1 locus in FY. Low polymorphism were observed at 1 locus in TBs and 2 loci in SWs. These results indicate that the breeding selection potential of these five genes was different in the four rabbit breeds. As a cultivated breed, the population genetic diversity of TBs is much higher than that of the other three indigenous breeds. These results are helpful to the effective exploration and conservation of the genetic resources for rabbit coat color.

The mammalian *MC1R* gene is a highly polymorphic gene. Its variations are linked with the alteration of mammalian coat or skin color, such as black rats [24], chestnut horses [25], red Holsteins [26], red and white pig [27], red and black goats [28], black chickens [29], and so on. In this study, two indels (c.284-285del and c.292-295del) were found in the TB and SW breeds, and polymorphism distribution of the c.292-295del sites were significantly different between TBs and SWs (*p* < 0.05). Previous investigations found that *MC1R* c.280_285del6 was more frequently presented in black rabbits, c.304_333del30 was recessive red/yellow, and c.[124A;125_130del6] was connected with Japanese brindling coat color [6,7,30]. The *MITF* gene polymorphism is known to be associated with the black and white colors in mice [31], pigs [32], cattle [33], horses [34], chickens [35], ducks [36], llamas [37], and so on. In this study, three SNPs (g.232587A < G, g.232650C < G, g.232766A < T) were found in the TB, SG, and SW breeds, and polymorphism distribution of the g.232587A < G sites were significantly different between TBs and SGs (*p* < 0.01). A previous study discovered that the mRNA expression levels of *MLPH* in the skin of black Rex rabbits was higher than that of other Rex rabbit skin types, and it was the lowest in the skin of white Rex rabbits [2]. *TYR* gene variation not only affects the occurrence of human albinism, but also has a significant correlation with animal coat color, body color, and meat color. Variations in the *TYR* gene cause melanin deficiency in human eyes, skin, and hair, resulting in albinism [38,39]. The mRNA expression of the *TYR* gene was significantly different in black and white sheep [40], dark-gray and light-gray goats [41], black and white feather ducks [42], as well as black and white feather chickens [43]. Additionally, there were significant differences in dark-gray and light-gray goat black fibers and different black-bone chicken muscles [41,42]. In this study, five SNPs (c.185G < A, c.465C < T, c.498T < C, c.669C < T, c.624C < T) were found in the TB, SG, and SW breeds. Polymorphism distribution of the c.185G < A site was significantly different between TBs and SGs (*p* < 0.01), and polymorphism distribution of the c.624C < T site was significantly different between TBs, SWs, and SGs (*p* < 0.01). In previous studies, a *TYR* homozygous SNP (T373K) in albino rabbits and a 3′UTR variation that targeted degradation of *TYR* mRNAs were identified [13]. The CRISPR/Cas9-mediated *TYR* knockout rabbit verified the function of these variations, which displayed albinism and graying phenotypes [18,44,45]. The *TYRP1* gene is another member of the tyrosinase gene family. It is widely reported to be involved in the formation of animal coat and skin color, including in pigs [46], cattle [47], sheep [48], goats [49], and minks [50]. In this study, one SNP (g.4137286G < A) was found in the TB, SG, SW, and FY breeds, and polymorphism distribution of this site was significantly different between the four breeds (*p* < 0.01). An SNP in exon 2 of the *TYRP1* gene was found in several rabbit breeds, which led to the early termination of translation of the *TYRP1* gene as a strong candidate for the rabbit brown coat color locus [14]. Variations in the *MLPH* gene cause melanin transport defects, resulting in dilution of coat colors in cats [51], dogs [52], cattle [53], chickens [54], sheep [55], and minks [56]. In this study, three SNPs (c.693C < G, c.851A < G, c.911G < A) were found in the TB and SW breeds. Polymorphism distribution of these three sites were significantly different between TBs and SGs (*p* < 0.01 or *p* < 0.05). In previous studies, two variations in the coding region of the *MLPH* gene, g.549853delG and c.111-5C > A, were significantly associated with rabbit coat color dilution traits [15,57,58].

## 5. Conclusions

In summary, five coat color-related genes have different variation sites in four Chinese native rabbit breeds. These loci have low-to-moderate polymorphism, and there are significant differences in their distribution among different breeds. The results of this study will play an important role in further investigation of the molecular mechanism of the regulation of coat color phenotype of Chinese native rabbit breeds and their high-quality genetic resources.

## Figures and Tables

**Table 1 animals-11-00081-t001:** Information of primers used for PCR.

Primer Names	Primer Sequence (5′→3′)	Fragment Size (bp)	Tm (°C)
*MC1R*	F: GCTCCCTCATGCCACC R: GAACATGCGGACGTACAAAA	606	58 °C
*MITF*	F: TGTTACTAATAGCCCTTTCC R: GGACACTTCTTTACCCTAG	729	56 °C
*TYR*	F: GTGAACCAGAGGGAACAT R: AAAGTGAGGTAGGCAAGG	741	58 °C
*TYRP1*	F: TGCCATACCAGACCAAG R: CAATGACAAACTGAGGG	682	56 °C
*MLPH*	F: CCTCCCTCAGTGCCACCTCT R: GGTCCCTAACTCCCACTTGG	498	56 °C

**Table 2 animals-11-00081-t002:** Allelic and genotypic distribution of the *MC1R* gene in four rabbit breeds.

Breed	Locus	Genotype Frequency		PIC	*p* Value
		Normal	Indel		
TB	c.284-285del	12(0.5000)	12(0.5000)	0.3750	0.1351
SW		18(0.7500)	6(0.2500)	0.3046	
TB	c.292-295del	15(0.6250)	9(0.3750)	0.3589	0.03633
SW		22(0.9167)	2(0.0833)	0.1411	

Note: TB: Tianfu black rabbit; SW: Sichuan white rabbit; SG: Sichuan gray rabbit; FY: Fujian yellow rabbit; PIC: polymorphism information content. Normal is the reference genotype retrieved from NCBI.

**Table 3 animals-11-00081-t003:** Allelic and genotypic distribution of the *MITF* gene in four rabbit breeds.

Breed	Locus	Genotype Frequency			Allele Frequency		PIC	*p* Value
	g.232587 A < G	AA	AG	GG	A	G		
TB		7(0.2917)	10(0.4167)	7(0.2917)	0.5000	0.5000	0.3750	0.00349
SG		18(0.7500)	2(0.0833)	4(0.1667)	0.7917	0.2083	0.2755	
	g.232650C < G	CC	CG	GG	C	G		
TB		8(0.3333)	8(0.3333)	8(0.3333)	0.5000	0.5000	0.3750	
	g.232766A < T	AA	AT	TT	A	T		
SW		3(0.1250)	4(0.1667)	17(0.7083)	0.2083	0.7917	0.2755	

**Table 4 animals-11-00081-t004:** Allelic and genotypic distribution of the *TYR* gene in four rabbit breeds.

Breed	Locus	Genotype Frequency			Allele Frequency		PIC	*p* Value
	c.185G < A	GG	GA	AA	G	A		
TB		2(0.0833)	18(0.7500)	4(0.1667)	0.4583	0.5417	0.3733	0.003862
SG		0(0.000)	10(0.4167)	14(0.5833)	0.2083	0.7917	0.2755	
	c.465C < T	CC	CT	TT	C	T		
TB		2(0.0833)	18(0.7500)	4(0.1667)	0.4583	0.5417	0.3733	
	c.498T < C	TT	TC	CC	T	C		
TB		2(0.0435)	18(0.7826)	4(0.1739)	0.4583	0.5417	0.3733	
	c.669C < T	CC	CT	TT	C	T		
TB		4(0.1667)	16(0.6667)	4(0.1667)	0.5000	0.5000	0.3750	
	c.624C < T	GG	GA	AA	G	A		
TB		2(0.0833)	2(0.0833)	20(0.8333)	0.1250	0.8750	0.1948	9.57 × 10^−9^
SW		2(0.0833)	7(0.2917)	15(0.6250)	0.2292	0.7708	0.2909	
SG		19(0.7917)	1(0.0417)	4(0.1667)	0.8125	0.1875	0.2583	

**Table 5 animals-11-00081-t005:** Allelic and genotypic distribution of the *TYRP1* gene in four rabbit breeds.

Breed	Locus	Genotype Frequency			Allele Frequency		PIC	*p* Value
	g.4137286G < A	GG	GA	AA	G	A		
TB		6(0.2500)	10(0.4167)	8(0.3333)	0.4583	0.5417	0.3733	0.008868
SW		0(0.0000)	8(0.3333)	16(0.6667)	0.1667	0.8333	0.2392	
SG		4(0.1667)	11(0.4583)	9(0.3750)	0.3958	0.6042	0.3639	
FY		0(0.0000)	15(0.6250)	9(0.3750)	0.3125	0.6875	0.3374	

**Table 6 animals-11-00081-t006:** Allelic and genotypic distribution of the *MLPH* gene in four rabbit breeds.

Breed	Locus	Genotype Frequency		Allele Frequency			PIC	*p* Value
	c.693C < G	CC	CG	GG	C	G		
TB		8(0.3333)	1(0.0417)	15(0.6250)	0.3542	0.6458	0.3528	6.29 × 10^−5^
SW		6(0.2500)	14(0.5833)	4(0.1677)	0.5417	0.4583	0.3733	
	c.851A < G	AA	AG	GG	A	G		
TB		8(0.3333)	1(0.0417)	15(0.6250)	0.3542	0.6458	0.3528	0.02236
SW		8(0.3333)	8(0.3333)	8(0.3333)	0.5000	0.5000	0.3750	
	c.911G < A	GG	GA	AA	A	G		
TB		11(0.4533)	0(0.0000)	13(0.5417)	0.4583	0.5417	0.3733	0.000962
SW		8(0.3333)	10(0.4167)	6(0.2500)	0.5417	0.4583	0.3733	

## Data Availability

The data presented in this study are available on request from the corresponding author.

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
