# Peer review of "Analysis of MC1R, MITF, TYR, TYRP1, and MLPH Genes Polymorphism in Four Rabbit Breeds with Different Coat Colors"

_animals, 2021, doi:10.3390/ani11010081_

Round 1

Reviewer 1 Report

I regret to inform you that I propose major revision this article. The language throughout the manuscript is of a bad quality, the discussion is not well-written and most of the discussion parts should be moved to introduction, since they are not related to the results of the manuscript. Moreover, chi square test should not be used on frequencies, as well as sequencing of heterozygous samples without cloning usually affects the ability to detect the distinct alleles as well as the haplotypes, if more than one mutation are mapped in the same amplicon. Finally, the authors present a possible association between the mutations (usually called loci) and the black phenotype, which is not supported by any further analysis.

Author Response

Re: Thank you very much for your careful reading of our manuscript and valuable comments. We have combed through the grammar and spelling mistakes in the manuscript. In the discussion section, we checked carefully and retained only the discussions and literature related to the results. Chi-square test is a widely used hypothesis testing method, which is used in statistical inference of classified data, including chi-square test with two or more rates and correlation analysis of classified data. For example, it used in these two papers,"Characterization of a dopamine transporter polymorphism and behavior in Belgian Malinois" and "Prognostic Value of EEG Microstates in Acute Stroke". This conjecture was based on the reports in the previous literatures and the distribution differences of mutations in different coat color breeds in this paper. In order to identified the relationship between these mutations and coat color, a larger sample size of association analysis is needed.

Reviewer 2 Report

GENERAL COMMENT:

I consider this work is within the scope of “Animals”. It contains information useful in a field in which available information is scarce and of interest to improve knowledge on rabbit coat colour genetics. I indicate several suggestions to be considered by the authors to improve the manuscript. I indicate these recommendations below and in a commented PDF file I have uploaded.

SIMPLE SUMMARY:

Line 13: Insert space after commas in: “MC1R,MITF,TYR,TYRP1”.

Line 14: Correct English writing. "were" does not fit here. I suggest: "showed".

ABSTRACT:

Line 21: TB, SW, SG and FY are not generally and internationally known initials. Therefore, it is necessary to write it with the full words.

KEYWORDS:

Line 27: Insert space after comma where indicated.

MATERIALS AND METHODS:

Line 73: Correct “audlt” as “adult”.

Lines 74-75: Tianfu black, Sichuan white, Sichuan gray and Fujian yellow rabbit breeds are of local distribution and not internationally known. Therefore, in order for international readers can search information of these breeds, if interested, it is necessary to add here bibliographic references on these rabbit breeds.

Line 80: It is said that “The candidate mutation was selected from the former publication”. However, it is not clear what is that publication. Therefore, it is better to quote here this bibliographic reference.

Line 81: Write “Oryctolagus cuniculus” in italics.

Line 98: Put (Nei and Roychoudhury, 1974) bibliographic quote in the format requested by the journal: number between square brackets. Subsequently revise the correct numbering sequence of the bibliographic quotes in the manuscript.

Line 100: Include bibliographic citation for this statistical package.

REFERENCES SECTION:

In general terms this section adjust to style and format of the journal. However, it is possible to review it to better adjust it to the journal style. For example, scientific names of the animals must be in italics, journal titles must be abbreviated, etc. I have indicated some of them in the annotated version of the manuscript I have uploaded, but there are other issues to improve.

TABLES:

Tables need to be interpreted autonomously without reading the manuscript text. It implies that all non-standard initials (TB, SW, WT, MT, PIC, SG, FY) need to be explained in a footnote of each table.

Round 2

Reviewer 1 Report

Dear Authors

I have some important suggestion for your manuscript in order to be publishable. There are still many grammatical errors throughout the manuscript: 

p.ex. Page 1 line 32, Page 2 line 49, page 3 line 80, etc. 

Page 1 line 34: "378 genes and mutation sites" needs to be rephrased (mutations in 378 genes perhaps)

Page 1 line 39: missing reference

Page 2 line 46: missing reference

Page 2 line 55: "The mutation in exon 2 (g.41360196G> A) of rabbit 55 TYRP1 resulted in the early termination codon at position 190 of amino acid sequence (p.Trp190ter)" this is called a nonsense mutation, you can replace

Page 3 line 99: What package did you use for the analysis? Maybe needs a citation

Page 3 lines 113-114: The amplicons are not of the same length, you have to normalize the values using the PCR product length to find the most polymorphic fragment

Page 4 line 119 and throughout the MS: the terms loci and locus are not uses in an appropriate way. You can use mutation sites or just mention "indels", "SNPs" etc

Page 4 line 121 and throughout the MS: by "wild type" probably you mean the identical with the reference, so you can refer to this as reference and alternative (if there are two) or a phrase of your choice. Also, a dominant allele means something different in genetics, so you can replace with "more frequent, less frequent"

Tables' legends can be rephrased with "Allelic and genotypic distribution of X gene"

Lines 169-176: How did you conclude that the SNPs are associated with the phenotypes? Is this a result from a statistical test?

Lines 192-199: Not a Discussion, you can move them to Intro

Line 236: Polymorphism distribution instead of polymorphic

Line 240: "And then" has to be removed

The discussion has to be re-written and to be based on your results, which you will compare with other studies and discuss similar or different findings. 
